# Animal and Human Dirofilariasis in India and Sri Lanka: A Systematic Review and Meta-Analysis

**DOI:** 10.3390/ani13091551

**Published:** 2023-05-05

**Authors:** Sandani S. Thilakarathne, Nicholas K. Y. Yuen, Mohammad Mahmudul Hassan, Thishan C. Yahathugoda, Swaid Abdullah

**Affiliations:** 1Department of Parasitology, Faculty of Medicine, University of Ruhuna, Galle P.O. Box 70, Sri Lanka; 2School of Veterinary Science, The University of Queensland, Gatton, QLD 4343, Australia; 3Faculty of Veterinary Medicine, Chattogram Veterinary and Animal Sciences University, Chattogram 4225, Bangladesh

**Keywords:** heartworm, *D. immitis*, *D. repens*, parasites, One Health, vector-borne infectious disease, mosquito-borne

## Abstract

**Simple Summary:**

Dirofilariasis is caused by *Dirofilaria* spp. worm infections, transmitted by mosquitoes, and affects humans and animals worldwide. Often, infected animals show symptoms relating to the cardiopulmonary system (heart and lung) and subcutaneous tissue (eye and skin). This study assessed the current published data on the distribution and prevalence of dirofilariasis across Sri Lanka and India. This analysis found that almost all cases of human dirofilariasis reported in Sri Lanka and India are presented as subcutaneous infections, with the eye being the most commonly affected organ. Both heartworm and subcutaneous infections are found in the dog populations in India. However, only subcutaneous infections have so far been reported in Sri Lanka, and the rationale behind this geographical distribution of infection patterns of dirofilariasis remains unknown and warrants further research. There was a low infection rate in the pet and working dog populations in India and Sri Lanka, but this may change due to climate change and emerging anti-parasitic drug resistance. It was identified in this study that some regions within India and Sri Lanka have not yet been surveyed for dirofilariasis, and future studies need to target these unsurveyed areas to better understand the geographical and species distribution of dirofilariasis in these two countries.

**Abstract:**

Dirofilariasis is an emerging vector-borne tropical disease of public health importance that mainly affects humans and dogs. *Dirofilaria immitis* and *D. repens* are the two well-documented dirofilariasis-causing filarioid helminths of both medical and veterinary concerns in India and Sri Lanka. This systematic review and meta-analysis aimed to describe and summarize the current evidence of dirofilariasis prevalence and distribution in India and Sri Lanka. Interestingly, *D. repens* is reported to circulate in both dogs (prevalence of 35.8% (95% CI: 11.23–60.69)) and humans (97% of published case reports) in India and Sri Lanka, but *D. immitis* is reported to be present in the dog populations in India (prevalence of 9.7% (95% CI: 8.5–11.0%)), and so far, it has not been reported in Sri Lanka. This peculiar distribution of *D. immitis* and *D. repens* in the two neighbouring countries could be due to the interaction between the two parasite species, which could affect the pattern of infection of the two worm species in dogs and thus influence the geographical distribution of these two filarial worms. In medical and veterinary practice, histopathology was the most commonly used diagnostic technique (31.3%; 95% CI 2.5–60.2%). The low specificity of histopathology to speciate the various *Dirofilaria* spp. may lead to misdiagnosis. It was identified in this study that several regions of India and Sri Lanka have not yet been surveyed for dirofilariasis. This limits our understanding of the geographical distribution and interspecies interactions of the two parasites within these countries. Parasite distribution, disease prevalence, and interspecies interactions between the vectors and the host should be targeted for future research.

## 1. Introduction

The dynamics of vector-borne disease transmission are often multifactorial. With climate change, unpredictable weather conditions can affect the survival and distribution of the vector populations and hence the diseases they transmit. Examples include the Dengue virus transmitted by mosquitoes [1], the Hendra virus transmitted by bats [2], and *Babesia* spp. transmitted by ticks [3]. Mosquito-borne diseases are arguably the most researched area in the field of vector-borne diseases. Mosquitoes can be readily transferred from one place to another by wind, air transport, or cargo ships, which then introduce and establish diseases and potential outbreaks to previously exotic/non-endemic areas. In addition, deforestation, urbanization, irrigation, and change in socio-economic circumstances have been implicated as factors contributing to the rise in mosquito-borne diseases, among other vector-borne diseases [4].

Dirofilariasis is an emerging zoonosis globally caused by infection with filarial nematodes of the genus *Dirofilaria*. Climate change may contribute to the changes in its temporal and spatial distribution and a possible increase in the incidence of this disease [5]. However, our understanding of this disease is limited and is often overshadowed by the more prominent mosquito-borne diseases, such as malaria, dengue, chikungunya, and West Nile fever.

Filarial worms of the genus *Dirofilaria* are long tubular adult worms which infect the circulatory system or connective tissues of mammalian hosts, and mature female worms produce unsheathed larvae called microfilariae, which circulate in the host blood. So far, around 34 *Dirofilaria* spp. have been described predominantly from carnivores and primates and allocated to two subgenera: *Dirofilaria*, comprising long thin worms (<0.1 mm diameter) with smooth cuticles that infect the lungs, heart, and blood vessels; *Nochtiella*, which are shorter stout worms (0.2–0.5 mm diameter) with cuticular ornamentations (longitudinal ridges with transverse striations) and infect subcutaneous and conjunctival tissues [6] (Table 1).

The two major species that are of both medical and veterinary concern are *Dirofilaria immitis* and *D. repens.* Infected humans and animals often share similar clinical signs, and the transmission cycles of both parasites are similar up to the point of larval migration in the host. The transmission cycle (Figure 1) begins when a competent vector (mosquito) ingests a blood meal from a *Dirofilaria*-infected definitive host (e.g., dogs) with microfilaria in its blood. The ingested microfilaria then matures from larval stage 1 to 3 (L1–L3) in the haemocoel of the infected mosquito vector in 2 to 3 weeks and then migrates to the mouth parts of this mosquito. The infective L3 larvae are then transmitted to a definitive host during the bite of this mosquito. *Dirofilaria immitis* larvae then develop from L3 to L4 in 3 to 12 days in the somatic tissues (e.g., muscles) of the mammalian host and later migrate into the bloodstream to reach the right ventricle and pulmonary arteries, where they moult from the L4 larvae to L5 immature adults, and they mature into adult heartworm in the subsequent 2–3 months. In contrast to *D. immitis*, *D. repens* L4 larvae do not migrate but moult and mature into an adult within the subcutaneous tissues. Once mature, adult worms of both species reproduce and produce microfilariae (pre-L1), which are released in the host blood vessels and taken up by a feeding mosquito [7].

While dogs and cats are well recognized as the definitive host, *D. repens* have, incidentally, been found in other animal species, such as racoons and monkeys, which are considered accidental hosts [8]. Many mosquito species have been identified as competent vectors, such as *Armigeres* spp., *Mansonia* spp., *Culex* spp., and *Aedes* spp. [9].

The diagnosis of infection is based on specific diagnostic tests. Commercially available veterinary serological test kits based on enzyme-linked immunosorbent assay (ELISA) and immuno-chromatographic test (ICT) can be used for antigen detection of female *D. immitis* in dogs, and microscopic examination of blood for microfilariae using Knott’s technique is a commonly used modality to identify a patent infection [10]. For *D. repens*, a blood screening using Knott’s technique is the most commonly used test [5]. In human medicine, radiographic imaging is primarily used to guide the diagnosis of *D. immitis*, including “coin lesions” revealed on chest X-rays and worm-like morphology on computed tomography (CT) and magnetic resonance imaging (MRI) scans, which is often followed by histological identification of larvae (often disintegrating) on nodule aspirates or excisional biopsies of the affected areas [11,12,13,14]. For *D. repens*, adult worms in subcutaneous or ocular nodules are extracted by excisional biopsies followed by morphological identification [15].

Treatments for adult *D. immitis* worm infections in dogs and cats are often complex due to potentially severe complications. One of the major concerns following adulticidal treatment is the sudden uncontrolled worm death post-treatment, which can cause a surge in cytokine release into the bloodstream leading to shock, and a high worm load can be fatal. A high amount of dirofilarial worms can also cause thrombo-embolism in the pulmonary arteries and veins, leading to vascular congestion and hypoxia. Ivermectin is mostly used as a monthly preventative to control heartworm infection in dogs, and melarsomine, in combination with doxycycline, is used as adulticide treatment in infected dogs, with a reported efficacy of 73% [16,17]. It is worth noting that the use of doxycycline (antimicrobial) is primarily to eliminate the *Wolbachia* bacteria, an endosymbiont of filarial worm, thereby disrupting the life cycle of filarial worms in various developmental stages [18,19]. It appears that doxycycline/ivermectin combination achieves a greater adulticidal effect, possibly by causing adult worm sterility [18,19]. Prednisolone is often used in conjunction to reduce inflammation and minimize the host immune response to the cytokine release associated with the treatment [20]. Although little information is available on the treatment of *D. repens* in animals, surgical removal of the nodule remains the treatment of choice in human medicine. Albendazole, diethylcarbamazine, and ivermectin are recommended to prevent their recurrence [21,22].

Published studies of *D. immitis* in dogs identified that dogs aged 5–6 years and male dogs are more likely to be diagnosed with heartworm infection [23,24]. While other animal species, such as domestic cats, ferrets, and wildlife (such as monkeys and wild canids), have also been implicated as susceptible hosts, the risk factors for *D. immitis* infection in these species are less frequently investigated [25,26,27]. *Dirofilaria repens* infections are more commonly reported in children and males [28]. Based on our literature search, little is known about the risk factors associated with *D. repens* infection in animals and *D. immitis* infection in humans.

While only some *Dirofilaria* spp. are zoonotic, it should be recognized as a public health risk, especially in parts of the world harbouring mosquito vectors and susceptible host populations. Vector-borne disease dynamics vary between geographical regions and are often driven by local geography and demography [29]. This dynamic is likely to shift unpredictably in certain areas due to climate change. Therefore, we need to widen and deepen our understanding of the distributions of dirofilariasis, particularly in the various regions of Asia, as these are arguably the most affected regions in the world [30].

The prevalence of dirofilariasis varies In different regions of the globe as vector and host distributions vary in different eco-climatic zones (Table 2). Among south Asian countries, Sri Lanka is the most affected, followed by India [6,31,32,33,34,35,36]. Thus, these two neighbouring countries, India and Sri Lanka, can be the appropriate starting point to review our current understanding of the epidemiology of dirofilariasis [6,31,32,33,34,35,36]. Interestingly, although they are neighbouring countries within the same eco-climatic zone and harbour suitable host and vector species, so far, Sri Lanka has only recorded cases of *D. repens* infection, while India has reported the presence of both *D. repens and D. immitis*. It remains unknown why these kinds of distribution variations exist between these two countries.

In this manuscript, we performed a systematic review and meta-analysis that summarizes the current understanding and prevalence of dirofilariasis in India and Sri Lanka; the review identifies and compares the prevalence of dirofilariasis between India and Sri Lanka, explores the potential reasons behind the differing prevalence in these neighbouring countries, and suggests the improvement of study designs for future investigations.

## 2. Materials and Methods

This systematic review was performed according to the Preferred Reporting Items for Systematic Review and Meta-analysis (PRISMA) guidelines [49] (Figure 2).

### 2.1. Strategy of Literature Search

A structured search of electronic databases, namely, PubMed, Scopus, and Web of Science, was performed until 22 February 2022, with no lower limit set to the date of publication. Search terms included pathogens (i.e., “*Dirofilaria repens*” OR “*Dirofilaria immitis*” OR “heartworm”) and place (i.e., “India” OR “Sri Lanka”). Bibliographies of review articles were regularly screened for potentially relevant articles and case reports that did not appear in the initial search.

### 2.2. Inclusion/Exclusion Criteria and Data Extraction

All article types, except original research articles, case reports, and review articles, were excluded. Published articles where the study population did not include humans or dogs were also excluded. Only peer-reviewed articles reporting the prevalence or cases of dirofilariasis in India or Sri Lanka, written in English, with full text available, were included in this manuscript. All records were first screened by titles and abstracts; then, the full text was reviewed prior to final inclusion.

Duplicate records were removed prior to data extraction. For included articles, titles, authors, publication year, type of study, country/state, species, age, sex, parasite, site of infection, the diagnostic technique, and prevalence were extracted and recorded in Microsoft Excel Spreadsheet 2019. Within the dog dataset, dog type was included and categorized into a pet, work, stray, and mixed types. The mixed dog type category included the dogs where the original publication did not specify the types of dogs in their study or did not analyse their data based on dog types.

### 2.3. Data Analysis

Descriptive statistics were performed and reported as percentages, with a 95% confidence interval (CI) using software package STATA/IC-13.0 (Stata Corp, 4905 Lakeway Drive, College Station, TX, USA). Meta-analysis was performed on the dog data to investigate the crude pooled estimated prevalence of parasitic cases, 95% confidence interval (CI), and the *p*-value. The studies’ variables were evaluated using the chi-square test on Cochran’s Q statistics (with *p*-value) followed by *I*^2^ statistics to determine the degree of heterogeneity. The weights were chosen to reflect the amount of information each study contains. A random-effect meta-analysis was applied using the “metan” command specifying the random effects due to the high degree of heterogeneity (*I*^2^ > 75%) [50]. The outputs were illustrated using a forest plot.

## 3. Results

The initial search yielded 463 records, of which 113 were duplicates. Out of 357 screened articles, 262 papers were excluded based on titles and abstracts. After screening the full text of the remaining articles, 74 records remained (Figure 2), of which there were 13 (13.6%) relating to dogs and 61 (82.4%) relating to humans. Of the 13 articles relating to dogs, 11 (84.6%) were from India, and 2 (16.4%) were from Sri Lanka (Table 3). Of the 61 articles relating to humans, 49 (80.3%) were from India, with 92 reported cases, and 12 (19.7%) were from Sri Lanka, with 138 reported cases (Table 4).

### 3.1. Descriptive Statistics of Human Data

The following factors were extracted from articles relating to human *Dirofilaria* infection: sex, affected body system, diagnostic technique, and parasite species and life cycle (Table 5). Interestingly, *D. repens* attributed to almost all (97.4%; 224/230) reported human cases. However, the parasitic life stage was not reported in more than 50% of cases. Overall, the incidence rates between males and females were similar, 38.3% (88/230) and 32.2% (74/230), respectively, with the eye (ocular) being the most commonly affected body system at 30.1% (98/260), followed by reproductive organs at 8.8% (23/260) and the oral region at 7.7% (20/260). In terms of diagnostic techniques (in combination or alone), 46% (182/396) of cases were diagnosed with surgical excision, followed by histopathological examination (19.2%; 76/396), and microscopy (14.9%, 59/396).

### 3.2. Meta-Analysis of Dog Data

The overall estimated pooled prevalence of *D. repens* was 35.8% (95% CI: 11.2–60.4%, *p* < 0.001) with significant heterogeneity of *I*^2^ = 100% (*p* < 0.001), and *D. immitis* was 9.1% (95% CI: 4.3–13.9%; *p* < 0.001) with significant heterogeneity (*I*^2^ = 95.6%) (Figure 3). *D. immitis* data represent the prevalence in India, as no cases have been reported from Sri Lanka (Table 6).

On the country level, the overall estimated pooled prevalence of both parasites in India was 23.2% (95% CI: 46.8–84.8%; *p* < 0.001) with significant heterogeneity (*I*^2^ = 99.9%), and in Sri Lanka, it was 49.0% (95% CI: 36.6–61.5%; *p* < 0.001) with significant heterogeneity (*I*^2^ = 71.4%) (Figure 4). Only *D. repens*, not *D. immitis,* has been reported in Sri Lanka.

The diagnostic techniques used for the detection of infection were compared. The majority of infections with both the parasite species were identified by histopathology using hematoxylin and eosin staining (HES; 31.3%; 95% CI: 2.5–60.2%; *p* < 0.001; *I*^2^ = 99.2%), followed by techniques that require microscopy, such as wet film, direct smear, and Knott’s test (21.6%; 95% CI 21.4–64.5%; *p* < 0.001; *I*^2^ = 100), and polymerase chain reaction (PCR; 20.2%; 95% CI 6.4–34.1%; *p* < 0.001; *I*^2^ = 98.0) (Figure 5).

In terms of dog types (i.e., pet, work, stray, or mixed), the overall estimated pooled prevalence of both the parasite species was highest in mixed dog types (35.5%; 95% CI 7.81–77.7%; *p* < 0.001; *I*^2^ = 100), followed by stray dogs (24.65%; 95% CI 12.0–37.3%; *p* < 0.001; *I*^2^ = 94.5) and working dogs (10.3%; 95% CI 2.7–17.8%; *p* < 0.001; *I*^2^ = 75.5) (Figure 6).

## 4. Discussion

To the best of the authors’ knowledge, this systematic review and meta-analysis represent the first study that summarized the published evidence of *Dirofilaria* spp. infections in humans and dogs in India and Sri Lanka. The results revealed that *D. repens* is the dominant *Dirofilaria* species infecting the dog and human populations in Sri Lanka. In India, a similar prevalence of *D. immitis* and *D. repens* was observed in the dog population, and *D. repens* remains the dominant species being reported to affect humans. In addition, traditional microscopy was the most commonly used diagnostic method in both the counties. Surprisingly, several regions of India and Sri Lanka have not yet been surveyed for dirofilariasis.

The analysis indicates that the prevalence of dirofilariasis in dogs in Sri Lanka is more than twice that of India, with *D. repens* being the sole *Dirofilaria* spp. infecting dogs in Sri Lanka, as compared to both *D. immitis* and *D. repens* in India. A recent report from the Eastern Province of Sri Lanka confirmed our finding, where almost 60% of dog samples tested positive for *D. repens*, and none were found positive for *D. immitis* [117]. While both *Dirofilaria* spp. are prevalent in India, it is interesting to note that the overall prevalence of dirofilariasis in India is lower than in Sri Lanka. However, this apparent difference may be affected or misrepresented by the random/inconsistent distributions and frequencies of surveyed areas in the two countries.

Human reports of dirofilariasis in both countries were almost exclusively due to *D. repens* (>97%), which is possibly true given the fact that *D. repens* can develop to adults in humans [92,118,119] and can occasionally produce microfilariae [120]. On the other hand, humans are suboptimal hosts for *D. immitis*; as such, the chances of finding the microfilaria in peripheral circulation are negligible [120], which makes diagnosis more difficult. It should be noted that in human medicine, most diagnoses are determined using the histological examination of surgical excisional biopsies, which does not facilitate accurate speciation of *Dirofilaria* spp. Employing more sensitive molecular assays for human testing could improve the detection of *D. immitis* infection [121] and improve our understanding of human dirofilariasis.

Another plausible explanation for the higher prevalence of *D. repens* in the host populations compared to *D. immitis* in the two countries is that *D. repens* is more competent in survival and persistence in the vector and/or the host, hence causing more disease in dogs and humans than *D. immitis* [122]; however, the underlying mechanism remains unknown. A similar phenomenon, termed “viral interference”, has been observed and well described in flaviviral infections in mosquitoes, whereby the presence of one flavivirus suppresses the replication of other flaviviruses in the mosquito [123]. From a preliminary study in 1995 involving experimental infection in dogs, it appears that *D. repens* may be the dominating species over *D. immitis*. [122], possibly explaining the higher prevalence of *D. repens* in both the human and dog populations. The same study also found that re-infection of the host with the same *Dirofilaria* spp. within 30 days would reduce the parasite burden within the host [122].

Further studies are required to (1) determine the underlying mechanism of the interaction between *D. repens* and *D. immitis* within the host and (2) determine whether various *Dirofilaria* spp. compete within the vector (mosquito), as well as the outcome of such competition, if any. The authors hypothesize that interaction between multiple *Dirofilaria* spp., such as *D. immitis* and *D. repens* in the vector, may lead to interference in the survival and transmission of these pathogens and the resulting prevalence in dogs and humans; however, this needs further investigation.

The analysis indicates that stray dogs were more likely to be tested positive for dirofilariasis. This is likely because these animals roam unrestricted and are not on any preventatives against the parasites, which may increase the chance of these animals being bitten by an infected mosquito. Pet dogs, however, have restricted movement and are usually on parasite preventatives. It is interesting to find that the prevalence of dirofilariasis in working dogs is similar to that of pet dogs, even though the working dogs spend most of the time outdoors in mosquito-prone areas; one possible explanation for this finding is that both pet and working dogs are more likely to receive prophylactic worm treatment. In addition, mosquitoes prefer to feed on resting subjects/animals where there is minimal disruption from the environment or the host during a blood meal. The results from the mixed dog types should be interpreted with care as the proportion of the various dog types are unknown, and results may be skewed towards certain dog types.

Amongst all body systems, the ocular system was found to be the most common site of infection in humans, likely due to mosquitoes’ ease of access to the peri-orbital areas [124]. Unfortunately, due to the nature of the data extracted from the case reports, it was impossible to fit the data into a multivariant regression model for a risk factor analysis. Further, while the risk factors considered in this study are broad and generic, not all studies reported these risk factors, restricting our ability to develop a robust model. However, it is worth noting that the majority of the cases being reported were from Kerala state in India and the central province in Sri Lanka. This indicates that there is a potentially higher risk of contracting *D. repens* infection in these regions of India and Sri Lanka. Moreover, the possibility that under-reporting or under-diagnosis of dirofilariasis in other regions of India and Sri Lanka leads to the apparent increase in cases in Kerala in India and the Central Province of Sri Lanka cannot be excluded.

The analysis indicates that surgical excision followed by histopathology was the most common diagnostic technique for diagnosing human dirofilariasis. However, histopathology alone is less likely to clearly distinguish between *D. immitis* and *D. repens* infection without some underlying assumptions, such as the location of infection. While *D. immitis* is known to reside in the cardio-pulmonary system, and *D. repens* resides in the dermatological or subcutaneous tissues, these parasites have been found outside of these common locations. For example, *D. immitis* has been found in the eye [121,125], reproductive system [126], and gastrointestinal system [127,128]. This could have led to inaccurate published results, which could have affected the results presented in this review. Therefore, the authors suggest that all histopathology-positive results, such as dirofilariasis, should be further characterized by molecular assays using species-specific primers that are more sensitive, specific, and accurate. While microscopy can also provide a definitive diagnosis with confidence, one would require extensive experience to differentiate the various types of *Dirofilaria* spp.

High heterogeneity has been identified in the meta-analyses of the dog population. This implies that results from various studies vary widely. This could be explained by (1) the various diagnostic techniques used in different studies and (2) the locations of the survey area. Different diagnostic techniques have different sensitivity and specificity, which would affect the prevalence of disease in certain areas. Moreover, the majority of the dog data came from two states/provinces in India/Sri Lanka, namely, Kerala and Assam in India and Western and North-Western provinces in Sri Lanka. It is known that microclimates in the various regions within a country affect mosquito, host, and pathogen distribution, survival, and dispersal, hence the varying degree of disease dynamics and prevalence in different areas. This signifies the need to perform region-based surveillance to monitor dirofilariasis prevalence with standardized diagnostic protocols and techniques, especially in the less-surveyed northern regions of India and Sri Lanka. Longitudinal vector-based studies identifying the variability or similarity of mosquito species and distributions across India and Sri Lanka would provide further understanding of the vector and disease dynamics.

## 5. Conclusions

India and Sri Lanka, situated in the humid tropical ecoclimatic zone with monsoonal weather patterns, favour the survival of mosquitoes and, thus, the transmission of mosquito-borne diseases. With global warming, climate change, and the emergence of anthelmintic resistance, extreme weather events are likely to become more frequent. The authors hypothesize that dirofilariasis is likely to become more prevalent in India and Sri Lanka, with the potential for the emergence of unidentified *Dirofilaria* spp. or the introduction of exotic species, as demonstrated recently in Tamil Nadu [129]. Future studies of passive and targeted surveillance should be designed carefully to provide meaningful conclusive results to inform public health measures, facilitate the comparison of results across various studies, and support the monitoring of changes over time.

## Figures and Tables

**Figure 1 animals-13-01551-f001:**
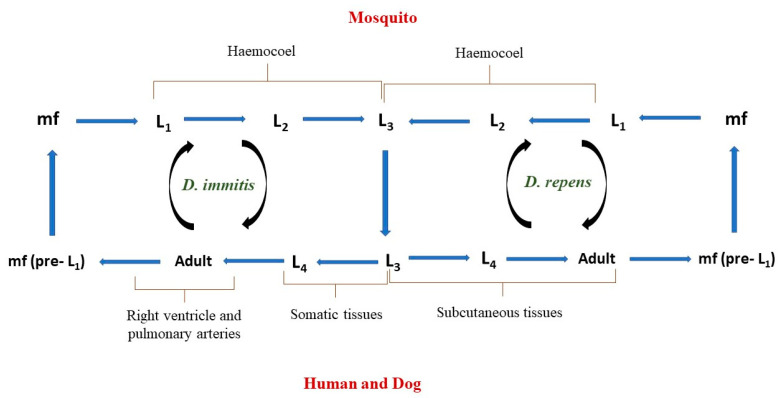
Life cycles of *D. immitis* and *D. repens*.

**Figure 2 animals-13-01551-f002:**
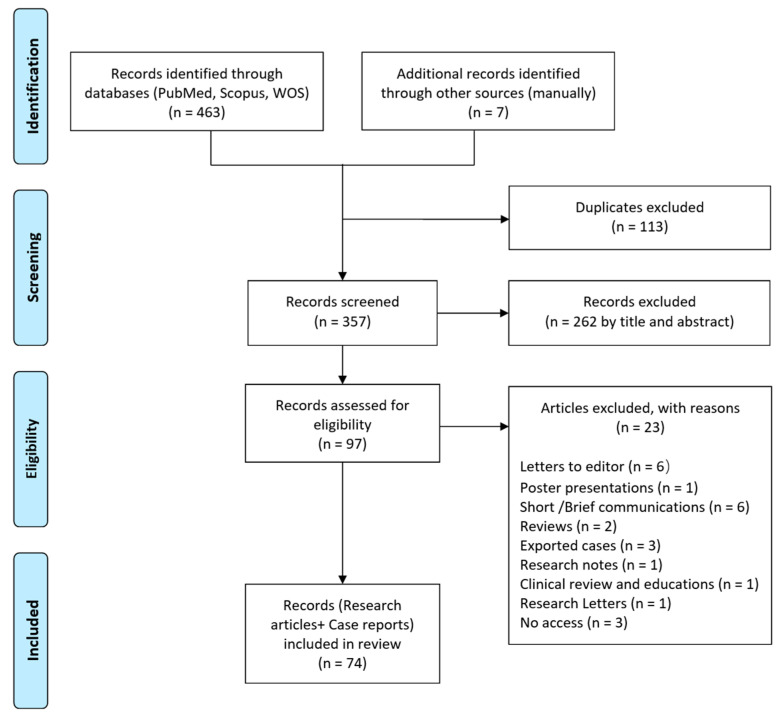
PRISMA flow diagram of the study selection process.

**Figure 3 animals-13-01551-f003:**
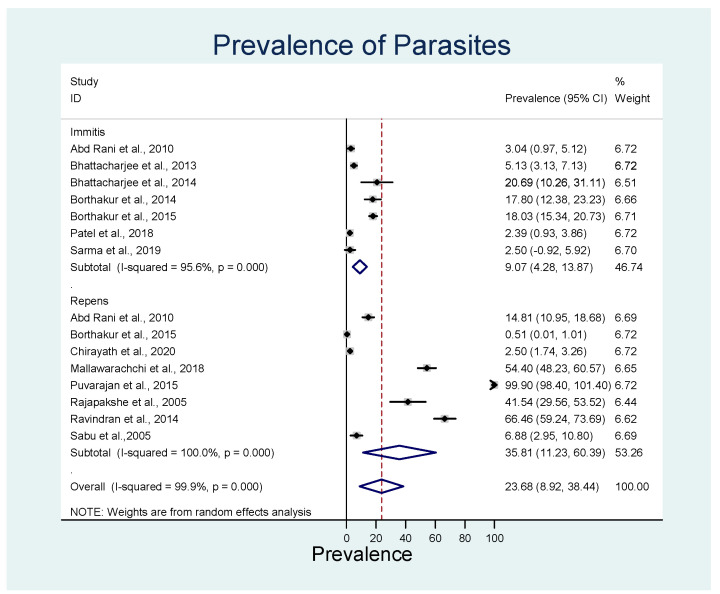
Forest plot of the pooled estimated prevalence of *D. repens* and *D. immitis*. The central square represents point estimates, whereas the square size represents the weight of each study in the meta-analysis. Diamonds represent the overall or summary effect for the respective category. Immitis = *D. immitis*; Repens = *D. repens* [45,51,52,53,54,55,56,57,58,59,60,61,62].

**Figure 4 animals-13-01551-f004:**
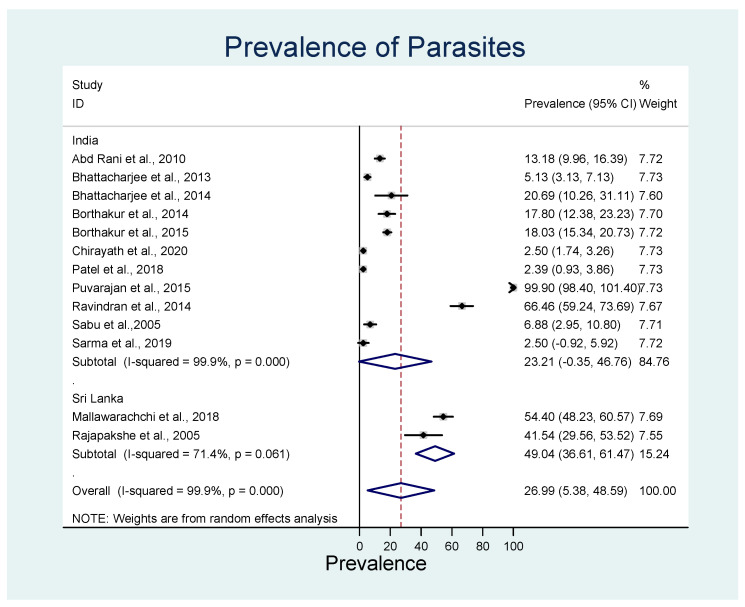
Forest plot of the pooled estimated prevalence of *D. repens* and *D. immitis* in India and Sri Lanka. The central square represents point estimates, whereas the square size represents the weight of each study in the meta-analysis. Diamonds represent the overall or summary effect for the respective category [45,51,52,53,54,55,56,57,58,59,60,61,62].

**Figure 5 animals-13-01551-f005:**
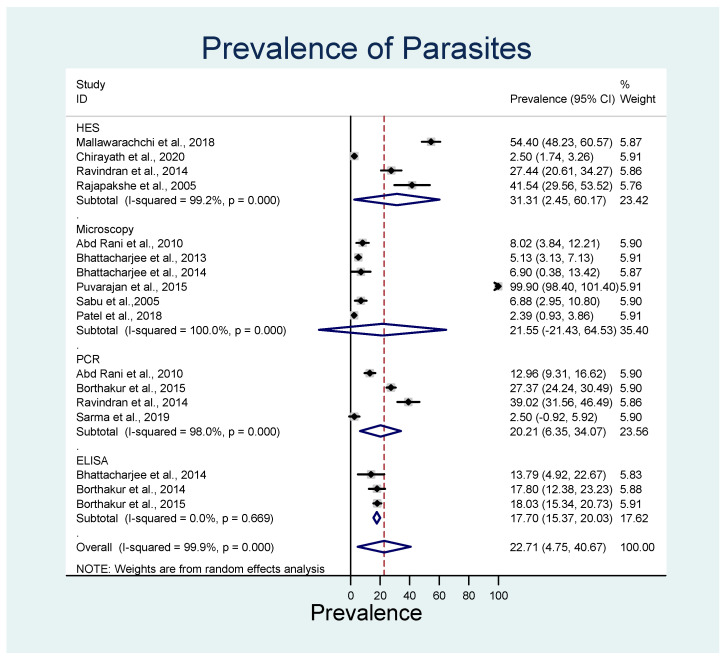
Forest plot of the pooled estimated prevalence of *D. repens* and *D. immitis* identified using different diagnostic techniques. The central square represents point estimates, whereas the square size represents the weight of each study in the meta-analysis. Diamonds represent the overall or summary effect for the respective category [45,51,52,53,54,55,56,57,58,59,60,61,62].

**Figure 6 animals-13-01551-f006:**
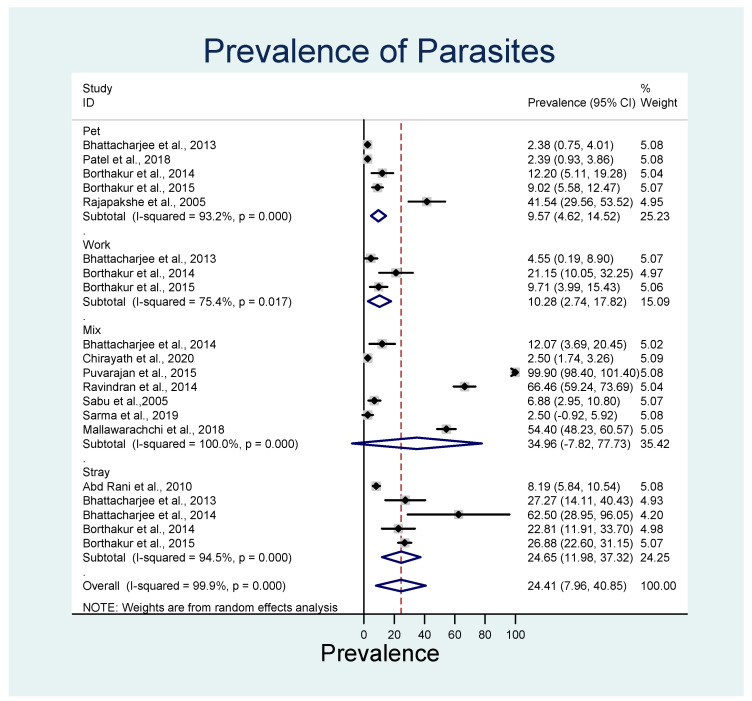
Forest plot of the pooled estimated prevalence of *D. repens* and *D. immitis* identified in different dog types. The central square represents point estimates, whereas the square size represents the weight of each study in the meta-analysis. Diamonds represent the overall or summary effect for the respective category [45,51,52,53,54,55,56,57,58,59,60,61,62].

**Table 1 animals-13-01551-t001:** Predilection sites of selected *Dirofilaria* spp. mainly found in India and Sri Lanka.

Parasite	Subgenus	Definitive/Incidental Host	Primary Site of Infection	Zoonosis	Distribution
*D. immitis*	*Dirofilaria*	Canid, felid, humans ^1^	Cardiovascular, Pulmonary	Yes	Americas, Europe, Indochina, Asia, Australia
*D. repens*	*Nochtiella*	Canid, felid, humans ^1^	Subcutaneous, subconjunctival, intermuscular tissues	Yes	Eurasia, Africa
*D. indica*	*Dirofilaria*	Dog	Heart	No	India
*D. linstowi*	*Nochtiella*	Primates	Subcutaneous tissues	No	Sri Lanka
*D. macae*	*Nochtiella*	Primates	Subcutaneous tissues	No	Indochina
*D. pagumae*	*Nochtiella*	Viverrid	Subcutaneous tissues	No	Indochina
*D. hongkongnesis*	*Nochtiella*	Canids, humans	Subcutaneous tissues	Yes	India, Hong Kong

^1^ Many definitive/incidental hosts (list not exhaustive), including wildlife.

**Table 2 animals-13-01551-t002:** Prevalence (last reported) summary of *D. immitis* and *D. repens* in various countries.

Countries	Regions/Areas Reported	Continent	Eco-Climatic Zone ^1^	Parasites	Last Reported Prevalence ^2^	Reference
Mediterranean countries	Canary Islands	Europe	Dry	*D. immitis*	22–40%	[37]
Romania	Southern	Europe	Dry/Humid temperate	*D. immitis*	5%	[38]
				*D. repens*	12%	
Slovakia	Trnava region	Central Europe	Humid temperate	*D. immitis*	64%	[39]
Bulgaria	N.A.	Central Europe	Humid temperate	*D. immitis*	34%	[40]
				*D. repens*	6%	
Russia	Southern and central areas	North Asia	Polar	*D. immitis*	36–55%	[41,42]
Iran	Kerman, southeastern	South Asia	Dry	*D. immitis*	5%	[43]
Tunisia	Northern and Central	Africa	Dry/Humid-temperate	*D. immitis*	15%	[44]
				*D. repens*	3%	
India	Assam	Southeast Asia	Humid tropical	*D. immitis*	30%	[34]
Sri Lanka	Western and North-western	Southeast Asia	Humid tropical	*D. repens*	69%	[45]
Thailand	Bangkok metropolitan region	Southeast Asia	Humid tropical	*D. immitis*	58%	[46]
Turkey	Sakarya, Kocaeli, Ankara, Elazig, and Mersin provinces	Western Asia	Dry/Humid-temperate	*D. immitis*	0–18%	[47]

N.A. = Not applicable. ^1^ Four ecoclimatic zones of the earth [48]. ^2^ Methodologies to determine prevalence of various parasites were different in different studies.

**Table 3 animals-13-01551-t003:** Geographical distribution of dog data.

Countries	State/Province ^1^	% (95%CI)	References
India (n = 454)	Kerala (n = 160)	35.24 (30.85–39.83)	[51,52,53]
	Assam (n = 157)	34.58 (30.21–39.16)	[54,55,56,57]
	Mizoram (n = 54)	11.89 (9.06–15.23)	[57,58]
	Maharashtra (n = 40)	8.81 (6.37–11.80)	[59]
	Tamil Nadu (n = 17)	3.74 (2.20–5.93)	[60]
	Delhi (n = 15)	3.30 (1.86–5.39)	[59]
	Goa (n = 10)	2.20 (1.06–4.01)	[61]
	Sikkim (n = 1)	0.22 (0.01–1.22)	[59]
Sri Lanka (n = 163)	Western Province (n = 114)	69.93 (62.27–76.86)	[45,62]
	Northwestern Province (n = 49)	30.06 (23.13–37.72)	[45]

^1^ State for India; Provinces for Sri Lanka.

**Table 4 animals-13-01551-t004:** Geographical distribution of human data.

Countries	State/Province ^1^	% (95%CI)	References
India (n = 92)	Kerala (n = 51)	55.43 (44.70–65.81)	[11,12,13,14,63,64,65,66,67,68,69,70,71,72,73,74,75,76,77,78]
	Karnataka (n = 18)	19.57 (12.03–29.15)	[79,80,81,82,83,84,85,86,87,88,89,90,91]
	Maharashtra (n = 9)	9.78 (4.57–17.76)	[92,93,94,95,96,97,98]
	Assam (n = 7)	7.61 (3.11–15.05)	[32,99,100]
	Thamil Nadu (n = 3)	3.26 (0.68–9.23)	[101,102]
	Bihar (n = 1)	1.09 (0.03–5.91)	[103]
	Delhi (n = 1)	1.09 (0.03–5.91)	[104]
	Goa (n = 1)	1.09 (0.03–5.91)	[105]
	Orissa (n = 1)	1.09 (0.03–5.91)	[82]
Sri Lanka (n = 138)	Central Province (n = 63)	45.65 (37.15–54.34)	[9,106,107,108,109,110]
	Western Province (n = 32)	23.18 (16.43–31.12)	[9,110,111,112,113,114,115]
	Uva Province (n = 26)	18.84 (12.69–26.37)	[116]
	Southern Province (n = 10)	7.24 (3.52–12.92)	[9]
	Sabaragamuwa Province (n = 3)	2.17 (0.45–6.22)	[9]
	Northwestern Province (n = 2)	1.44 (0.17–5.13)	[9]
	Eastern Province (n = 1)	0.72 (0.01–3.97)	[9]
	North Province (n = 1)	0.72 (0.01–3.97)	[9]

^1^ State for India; Provinces for Sri Lanka

**Table 5 animals-13-01551-t005:** Descriptive summary of human data on dirofilariasis in India and Sri Lanka.

Variables	Characteristics	% (95%CI)
Sex (n = 230)	Male (n = 88)	38.26 (31.95–44.87)
	Female (n = 74)	32.17 (26.18–38.63)
	N.A. (n = 68)	29.56 (23.74–35.91)
Diagnostic technique (n = 396)	Surgical excision (n = 182)	45.95 (40.97–51.00)
	Histopathological examination (n = 76)	19.19 (15.42–23.42)
	Microscopy (n = 59)	14.89 (11.53–18.79)
	Ultrasound (n = 38)	9.59 (6.88–12.93)
	Ophthalmic examination (n = 18)	4.54 (2.71–7.08)
	PCR (n = 11)	2.77 (1.39–4.91)
	Imaging (n = 6)	1.51 (0.55–3.26)
	FNA (n = 3)	0.75 (0.15–2.19)
	Self-emerged (n = 3)	0.75 (0.15–2.19)
Parasite species (n = 230)	*D. repens* (n = 224)	97.39 (94.41–99.04)
	*D. immitis* (n = 5)	2.17 (0.71–4.99)
	*Dirofilaria* spp. (n = 1)	0.43 (0.01–2.39)
Body system (n = 260)	Eye (n = 98)	30.05 (31.77–43.88)
	Repro (n = 23)	8.84 (5.69–12.97)
	Oral (n = 20)	7.69 (4.76–11.63)
	Hand (n = 18)	6.92 (4.15–10.72)
	Chest/Breast (n = 14)	5.38 (2.97–8.86)
	Abdomen (n = 13)	5.00 (2.68–8.39)
	Cheek/Face (n = 11)	4.23 (2.13–7.44)
	Neck (n = 11)	4.23 (2.13–7.44)
	Leg (n = 9)	3.46 (1.59–6.46)
	Head (n = 4)	1.53 (0.04–3.89)
	Respiratory (n = 2)	0.07 (0.09–2.75)
	Buttock (n = 1)	0.03 (0.09–2.12)
	Cardiovascular (n = 1)	0.03 (0.09–2.12)
	Digestive (n = 1)	0.03 (0.09–2.12)
	N.A. (n = 34)	13.07 (9.22–17.79)
Parasite life stage (n = 230)	Female (n = 39)	16.95 (12.34–22.44)
	Female adult (n = 14)	6.08 (3.36–10.00)
	Female immature (n = 10)	4.34 (2.10–7.85)
	Dead (n = 10)	4.34 (2.10–7.85)
	Male (n = 8)	3.47 (1.51–6.73)
	Live (n = 4)	1.73 (0.47–4.39)
	Degenerated (n = 3)	1.30 (0.26–3.76)
	Female gravid (n = 2)	0.86 (0.11–3.11)
	Female gravid with microfilaria (n = 1)	0.43 (0.01–2.39)
	Female dead (n = 1)	0.43 (0.01–2.39)
	Infertile (n = 1)	0.43 (0.01–2.39)
	Male adult (n = 1)	0.43 (0.01–2.39)
	Male immature (n = 1)	0.43 (0.01–2.39)
	Mature adult (n = 1)	0.43 (0.01–2.39)
	Microfilaria (n = 1)	0.43 (0.01–2.39)
	Worm fragmented (n = 1)	0.43 (0.01–2.39)
	N.A. (n = 132)	57.39 (50.72–63.86)

N.A. = Not applicable.

**Table 6 animals-13-01551-t006:** The overall prevalence of *D. immitis* and *D. repens* is categorized by country.

	India Prevalence (95% CI; Positive/Total)	Sri Lanka Prevalence (95% CI; Positive/Total)
*D. immitis*	9.7% (8.5–11.0%; 225/2318)	Nil
*D. repens*	8.1% (7.2–9.2%; 229/2814)	51.7% (46.1–57.4%; 163/315)

## Data Availability

Not applicable.

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
