# Peer review of "Animal and Human Dirofilariasis in India and Sri Lanka: A Systematic Review and Meta-Analysis"

_animals, 2023, doi:10.3390/ani13091551_

Round 1

Reviewer 1 Report

This is an interesting comparison of canine and human dinofilariasis in India and Sri Lanka. This disease is likely to increase due to climate change over the next number of years, and the review is a valuable addition to this information, even if the geographical area is limited. The manuscript is interesting, and well written, and I only have a few minor comments below

Line 21- working dogs rather than work dogs may sound better here

Line 99-100- it may be worth including the antigens in here which are used in the immunofluorescence and ELISA kits

Table 2- maybe worth including the methodologies in here

Line 195-196- not sure that this makes complete sense, so please reword

Table 3 and 4 - is it worth putting this on a map so it is more visual for the reader as my geography of India is not that good. Only a suggestion, but up to the authors

Line 218- were differences seen in the prevalence based on the different detection methods?

Line 308- is there any risk of recombination between the two species?

Line 315- you could delete ‘infected when’ here

Line 317- again working may sound better (on both occasions) than work

Line 319- same here with work too

Line 356- is there any similarities between the areas with the highest prevalence in terms of climate?

Author Response

This is an interesting comparison of canine and human dinofilariasis in India and Sri Lanka. This disease is likely to increase due to climate change over the next number of years, and the review is a valuable addition to this information, even if the geographical area is limited. The manuscript is interesting, and well written, and I only have a few minor comments below

The authors thank the reviewer for the positive comments.

Line 21- working dogs rather than work dogs may sound better here

Authors response: Thank you. Changed as suggested throughout the MS.

Line 99-100- it may be worth including the antigens in here which are used in the immunofluorescence and ELISA kits

Authors response: Thank you. Many different types of commercially available kits of various brands are available depending on regions and countries. The authors believe that by stating “antigen detection of female D. immitis” in line 101 – 102 has provided sufficient information in this context.

Table 2- maybe worth including the methodologies in here

Authors response: Thank you. The authors have stated in the footnote of the table that methodology used in each study varies.

Line 195-196- not sure that this makes complete sense, so please reword

Authors response: Thank you. Sentence removed. Instead, the authors specified that meta-analysis was performed on the dog data only. Line 191.

Table 3 and 4 - is it worth putting this on a map so it is more visual for the reader as my geography of India is not that good. Only a suggestion, but up to the authors

Authors response: We thank the reviewer’s suggestion. The authors explored this option. However, we believe that these data are best presented in a table format. As some states/provinces had very low case number or sample size, displaying the prevalence data on a map may be misleading.

Line 218- were differences seen in the prevalence based on the different detection methods?

Authors response: Meta-analysis and prevalence estimation were not able to be performed in the human data as all data were obtained from case reports. However, this analysis has been performed on the dog data and presented in Figure 5.

Line 308- is there any risk of recombination between the two species?

Authors response: D. immitis and D. repens are dirofilarial nematodes. Recombination between the two species is unlikely.

Line 315- you could delete ‘infected when’ here

Authors response: Thank you. Deleted as suggested. Line 321.

Line 317- again working may sound better (on both occasions) than work

Authors response: Thank you. Changed as suggested.

Line 319- same here with work too

Authors response: Thank you. Changed as suggested.

Line 356- is there any similarities between the areas with the highest prevalence in terms of climate?

Authors response: We did not compare the results based on the climate as it was out of our scope.

Reviewer 2 Report

Title: Animal and Human Dirofilariasis in India and Sri Lanka: A 2 Systematic Review and Meta-Analysis

With this study, a meta-analysis was applied to inform the status of dirofilariasis in humans and animals in India and Sri Lanka.

The selection of the data used in the study, the analysis applied, the writing of the article and the discussion are quite clear and enlightening.

In the introduction part of the article, the authors presented up-to-date data on the distribution, life cycle and pathogenesis and clinical manifestations of dirofilariasis.

In the material and method part, they analyzed the data they obtained as a result of scanning in different sources by following the PRISMA guide. The authors described their findings by showing the data they obtained on tables and figures, and in the discussion part, they compared their own data with the data of other researchers using the current literature.

Please find my comments on the manuscript also in attached file

Author Response

Title: Animal and Human Dirofilariasis in India and Sri Lanka: A 2 Systematic Review and Meta-Analysis

With this study, a meta-analysis was applied to inform the status of dirofilariasis in humans and animals in India and Sri Lanka.

The selection of the data used in the study, the analysis applied, the writing of the article and the discussion are quite clear and enlightening.

In the introduction part of the article, the authors presented up-to-date data on the distribution, life cycle and pathogenesis and clinical manifestations of dirofilariasis.

In the material and method part, they analyzed the data they obtained as a result of scanning in different sources by following the PRISMA guide. The authors described their findings by showing the data they obtained on tables and figures, and in the discussion part, they compared their own data with the data of other researchers using the current literature.

Please find my comments on the manuscript also in attached file

The authors would like to thank the reviewer for the positive feedback.

Line 115: Please add something about wolbachia bacteria here.

Authors response: Thank you for the suggestion. Added as suggested. Line 119 – 123.

Line 149: There are some data that published in Turkey by Simsek et al. Please use them in the table.

Authors response: Thank you. Added in Table 2.

Reviewer 3 Report

Line 21: working dogs

Line 65: are long nematodes

Table 1:  D. macae

D. repens in North America? Please, provide references

Lines 120-121: ? should be one sentence?

Line 126: what about free-living canids? Red fox, grey wolf, golden jackals (Alsarraf M. et al. 2023, Parasitol Res)

Table 2 is very fragmentary: does not report D. repens prevalence in Poland or Baltic countries, where recent studies were performed (Alsarraf et al. 2021). I am not convinced that this table is informative enough and not misleading; maybe it is better to cite one of the recent review articles , like Fuehrer HP et al. 2021, and provide only data for India and Sri Lanka.

Figure 2: imported cases

Line 181: working dogs (use through the text)

Line 189: ‘were calculated on different parasitic diseases between factors.’ ? not clear, clarify

Lines 212-219: different number of cases is presented (230, 260,396) and it is not clear for which group/subgroup refer these numbers; please expand and clarify.

Figure 3: should be: D. repens (not “Repens” ) and D.immitis

Table 6: what is Nil?

Check spelling of Knott through ms

Line 255, 258: working dog

Line 291: citation on ref 117 on Loa loa does not seem justified here; please cite Rossi et al. 2015 Genetic diversity of Dirofilaria spp. isolated from subcutaneous and ocular lesions of human patients in Ukraine, Acta Tropica

Lines 342-343; also Rossi et al. 2015 to be cited here

Author Response

Reviewer 3

The authors would like to thank the reviewer for the constructive feedback.

Line 21: working dogs

Authors response: Thank you. Changed throughout MS.

Line 65: are long nematodes

Authors response: Thank you. Changed as suggested. Line 66.

Table 1:  D. macae

Authors response: Thank you. Corrected.

  1. repens in North America? Please, provide references

Authors response: Thank you. The authors apologize for the mistake. This has now been removed.

Lines 120-121: ? should be one sentence?

Authors response: Thank you. Corrected. Line 126.

Line 126: what about free-living canids? Red fox, grey wolf, golden jackals (Alsarraf M. et al. 2023, Parasitol Res)

Authors response: Thank you. Added as suggested. Line 131 – 132.

Table 2 is very fragmentary: does not report D. repens prevalence in Poland or Baltic countries, where recent studies were performed (Alsarraf et al. 2021). I am not convinced that this table is informative enough and not misleading; maybe it is better to cite one of the recent review articles , like Fuehrer HP et al. 2021, and provide only data for India and Sri Lanka.

Authors response:

Figure 2: imported cases

Authors response: Thank you. The authors would appreciate some clarification on this comment.

Line 181: working dogs (use through the text)

Authors response: Thank you. Changed as suggested throughout MS.

Line 189: ‘were calculated on different parasitic diseases between factors.’ ? not clear, clarify

Authors response: Thank you. This sentence has been reworded. Line 193.

Lines 212-219: different number of cases is presented (230, 260,396) and it is not clear for which group/subgroup refer these numbers; please expand and clarify.

Authors response: Thank you. Not all data were available in the case reports. In some cases, more than one diagnostic technique was used, and more than one body system being affected. Details of groups/subgroups are presented in Table 5.

Figure 3: should be: D. repens (not “Repens” ) and D.immitis

Authors response: Thank you. A footnote has been added to explain the abbreviation used in the figure.

Table 6: what is Nil?

Authors response: Thank you. Of all the articles included in this systematic review, no study has been identified to have reported the prevalence or testing of D. immitis in Sri Lanka.

Check spelling of Knott through ms

Authors response: Thank you. Corrected throughout MS.

Line 255, 258: working dog

Authors response: Thank you. Corrected throughout MS.

Line 291: citation on ref 117 on Loa loa does not seem justified here; please cite Rossi et al. 2015 Genetic diversity of Dirofilaria spp. isolated from subcutaneous and ocular lesions of human patients in Ukraine, Acta Tropica

Authors response: Thank you. Changed as suggested.

Lines 342-343; also Rossi et al. 2015 to be cited here

Authors response: Thank you. Added as suggested.